# Facile Synthesis of Nitrogen-Rich Porous Carbon/NiMn Hybrids Using Efficient Water-Splitting Reaction

**DOI:** 10.3390/polym15143116

**Published:** 2023-07-21

**Authors:** Thirukumaran Periyasamy, Shakila Parveen Asrafali, Seong-Cheol Kim, Jaewoong Lee

**Affiliations:** 1Department of Fiber System Engineering, Yeungnam University, Gyeongsan 38541, Republic of Korea; 2School of Chemical Engineering, Yeungnam University, Gyeongsan 38541, Republic of Korea

**Keywords:** porous carbon, hetero-atom, bimetallic, electrocatalyst, water splitting

## Abstract

Proper design of multifunctional electrocatalyst that are abundantly available on earth, cost-effective and possess excellent activity and electrochemical stability towards oxygen evolution reaction (OER) and hydrogen evolution reaction (HER) are required for effective hydrogen generation from water-splitting reaction. In this context, the work herein reports the fabrication of nitrogen-rich porous carbon (NRPC) along with the inclusion of non-noble metal-based catalyst, adopting a simple and scalable methodology. NRPC containing nitrogen and oxygen atoms were synthesized from polybenzoxazine (Pbz) source, and non-noble metal(s) are inserted into the porous carbon surface using hydrothermal process. The structure formation and electrocatalytic activity of neat NRPC and monometallic and bimetallic inclusions (NRPC/Mn, NRPC/Ni and NRPC/NiMn) were analyzed using XRD, Raman, XPS, BET, SEM, TEM and electrochemical measurements. The formation of hierarchical 3D flower-like morphology for NRPC/NiMn was observed in SEM and TEM analyses. Especially, NRPC/NiMn proves to be an efficient electrocatalyst providing an overpotential of 370 mV towards OER and an overpotential of 136 mV towards HER. Moreover, it also shows a lowest Tafel slope of 64 mV dec^−1^ and exhibits excellent electrochemical stability up to 20 h. The synergistic effect produced by NRPC and bimetallic compounds increases the number of active sites at the electrode/electrolyte interface and thus speeds up the OER process.

## 1. Introduction

The depletion of fossil fuels have increased the demand for renewable energy in energy storage applications. In this context, the need for environmentally friendly, sustainable and highly efficient energy storage materials are much preferred. As water is available abundantly, splitting water using electrolysis to produce hydrogen is an effective way to produce sustainable energy [1,2,3,4]. The water-splitting reaction is a combination of hydrogen evolution reaction (HER) and oxygen evolution reaction (OER), in which OER is a very sluggish process that slows down the rate of water splitting. Until now, Pt/C, noble metal-based catalyst, has been effective in HER and precious metal oxides, such as IrO_2_ and RuO_2_, are effective in OER. Nevertheless, their scarcity and high cost hamper their usage on a large scale. Therefore, it is essential to design a cheap and eco-friendly electro catalyst that effectively boosts the reaction kinetics of HER and OER [5,6,7].

Modifications pertaining to hetero-atom-doped carbon and transition metals are the most feasible routes to utilize the full potential of a material’s catalytic activity. Hetero-atom-doped carbon can bring about efficient electron transfer through electronic structure regulation [8,9,10], whereas transition metal inclusion increases the interfacial interaction through vacancy modification. Instead of single metal, bimetallic oxides/hydroxides can produce several oxidation and reduction states that could facilitate the adsorption of hydroxide ions during water-splitting process [11,12,13]. A combination of hetero-atom-doped carbon and bimetallic oxides can bring out the synergistic effect of both of them, through the formation of both metal–oxide and metal–metal bonds. Moreover, the three-dimensional hierarchical structure produced from these composites enhances their catalytic activity [14,15,16,17,18,19,20,21].

Qin et al. synthesized two trifunctional electrocatalysts [1], NCNT/Ni-NiMn_2_O_4_ and NCNT/Ni-NiFe_2_O_4_, for HER and OER, using pyrolysis and found that they displayed smaller overpotential of 250 and 300 mV towards OER and 140 and 188 mV towards HER, respectively, at 10 mA cm^−1^. They also found that NCNT/NiMn_2_O_4_ displayed high specific capacity (502 mAh g^−1^) and power density (105 mW cm^−1^) comparable to Zn–air batteries. Hu et al. fabricated MoO_4_^2−^ (NiCo-LDH@PANI) and studied the effect of addition of MoO_4_^2−^ to NiCo-LDH@PANI^4^. They found that M-LDH@PANI-0.5 possesses a small overpotential of 266 mV at 10 mA cm^−1^ and acts as an effective catalyst for OER. Thangavel et al. fabricated a highly efficient electrocatalyst, MoNi_4_/MoO_2_@Ni, for OER, by immobilizing Ni_3_N particles on the surface of amorphous oxy-hydroxides [3]. The electrocatalyst possess a small overpotential of 271 mV at 10 mA cm^−1^, high turnover frequency of 2.53 s^−1^ and high Faradaic efficiency of 99.6%. Zhai et al. fabricated NiFe LDH containing both oxygen and metal vacancies [2]. They found that Ni_0.3_Fe_0.7_LDH@NF fabricated using this strategy has much lower overpotential of 184 mV at 10 mA cm^−1^, efficient for OER electrocatalysts.

Based on all the previous work, in the present work, we made an attempt to fabricate electrocatalyst that can bring out the synergistic effect of hetero-atom-doped carbon materials and bimetallic oxides. The hetero-atom-doped carbon was produced from a polybenzoxazine source, so that the resulting carbon is rich in nitrogen and oxygen atoms. The bimetallic oxide is preferred over monometallic as various oxidation/reduction states and increased interlayer spacing could be expected from these materials. A simple strategy of hetero-atom-doped carbon through polybenzoxazine calcination and bimetal inclusion using hydrothermal method has been adopted to fabricate the electrocatalysts. The preliminary structural characterizations of the synthesized materials, i.e., nitrogen-rich porous carbon (NRPC), monometal inclusion (NRPC/Mn and NRPC/Ni) and bimetal inclusion (NRPC/NiMn) and their electrocatalytic performance for HER and OER are investigated and discussed in detail.

## 2. Materials and Methods

Phenylethylamine, hydroquinone and paraformaldehyde were purchased from Sigma Aldrich (St. Louis, MO, USA). Potassium hydroxide (KOH), sodium hydroxide (NaOH), dimethyl sulfoxide (DMSO), nickel nitrate hexahydrate [Ni(NO_3_)_2_·6H_2_O], manganese nitrate tetrahydrate [Mn(NO_3_)_2_·4H_2_O], polyvinylidene fluoride (PVDF) and N, N-dimethylformamide (DMF) were purchased from Duksan Chemicals Co., Ltd. Gyeonggi-do Republic of Korea. All chemicals were used without further purification.

### 2.1. Synthesis of Hydroquinone-Phenylethylamine-Based Benzoxazine Monomer (HPh-Bzo)

The benzoxazine monomer was prepared through Mannich condensation using hydroquinone, phenylethylamine and paraformaldehyde as starting materials in 1:2:4 ratio, in presence of DMSO solvent. A little excess of paraformaldehyde was added to maintain the proper ratio (as some amount could be evaporated during synthesis). The reaction is performed for 5 h at 120 °C, resulting in the formation of pale yellow colored solution. The obtained reaction solution precipitates in 1 N sodium hydroxide solution, where the unreacted reactants dissolve in the solution. The precipitate after water washing and drying at 60 °C overnight, yields the product, i.e., HPh-Bzo (hydroquinone and phenylethylamine-based benzoxazine) with 75% yield.

### 2.2. Synthesis of Nitrogen-Rich Porous Carbon (NRPC)

The synthesized benzoxazine monomer (HPh-Bzo) was converted into NRPC via three different process, i.e., polymerization, carbonization and activation. HPh-Bzo undergoes thermal curing polymerization following a step-wise heating schedule of 100–250 °C, with an increment of 50 °C at each step along with a 1 h holding time. After this process, the benzoxazine monomer (HPh-Bzo) was converted into polybenzoxazine [poly(HPh-Pbz)]. Then, the polymer was converted into the carbon material through carbonization, by heating up to 600 °C, with a heating rate of 2 °C/min. After which, the carbonized material was mixed with twice the amount of aqueous KOH and kept overnight. It was then filtered, dried at 120 °C and activated by heating in a tubular furnace up to 800 °C, following a heating rate of 3 °C/min. Following activation, the products underwent multiple washings with 1 M HCl and deionized H_2_O until a neutral pH was achieved. Finally, the sample was dried at 110 °C for 12 h to obtain NRPC (nitrogen-rich porous carbon) with 42% yield. (Figure 1).

### 2.3. Synthesis of NRPC/Mn, NRPC/Ni and NRPC/NiMn

For synthesizing NRPC/Mn, a 250 mL of metal salt solution containing 6 mM of Mn(NO_3_)_2_·4H_2_O was prepared and the synthesized NRPC (10 mg) was dispersed into this solution. After which, NH_4_F (18 mM) was added to the dispersed solution and kept under ultra-sonication for 1 h. In the meantime, a basic solution containing NaOH and Na_2_CO_3_ in 1:2 ratio was prepared and added to the ultra-sonicated solution and stirred vigorously for 4 h. Then, the solution was aged overnight and sealed in an autoclave at 90 °C for 12 h. The obtained product was washed and dried completely to obtain NRPC/Mn. Similar procedure was followed to synthesize NRPC/Ni and NRPC/NiMn, where Ni(NO_3_)_2_·6H_2_O was used for synthesizing NRPC/Ni, and a mixture of Mn(NO_3_)_2_·4H_2_O and Ni(NO_3_)_2_·6H_2_O were used for synthesizing NRPC/NiMn (Figure 2). The obtained materials, i.e., NRPC, NRPC/Mn, NRPC/Ni and NRPC/NiMn were analyzed for preliminary characterizations and electrochemical performance.

### 2.4. Electrochemical Measurements

Electrochemical measurements were carried out using the CS2350 electrochemical workstation in a three-electrode system at room temperature for both HER and OER. Ni foam/NRPC, NRPC/Mn NRPC/Ni and NRPC/NiMn were used as the working electrode, a standard Hg/HgO electrode was used as the reference electrode and a Pt foil was used as the counter electrode. Furthermore, a 1 M aq. KOH solution was used as the electrolyte. For OER measurements, the system was saturated with O_2_, by purging high-purity oxygen and maintaining the bubbling throughout the measurement. Cyclic voltammetry (CV) measurements were conducted at different scanning rates from 5–100 mV s^−1^. Linear sweep voltammetry (LSV) was performed at a constant rate of 5 mV s^−1^. Tafel plots were fitted by considering the linear part of overpotential versus current density in log scale (log |j|), from which the Tafel slopes were calculated. All the potentials were calibrated versus reversible hydrogen electrode (RHE) using the equation given below:E_RHE_ = E_Hg/HgO_ + 0.0592 pH + 0.098 V.(1)

The pH of the electrolyte used was found to be 13.4.

## 3. Instrumentation

The prepared materials, i.e., NRPC, NRPC/Mn, NRPC/Ni and NRPC/NiMn were characterized by various physicochemical techniques such as field emission scanning electron microscopy (FESEM) with energy-dispersive X-ray spectroscopy (EDS), high-resolution transmittance electron microscopy (HRTEM), X-ray diffraction (XRD), Raman spectroscopy, nitrogen adsorption/desorption isotherms and X-ray photoelectron spectroscopy (XPS). FESEM was carried out on a Hitachi S-4800 equipped with EDX at an accelerating voltage of 4 kV. TEM/HRTEM images were performed with an FEI-Tecnai TF-20 transmission electron microscope with an operating accelerating voltage of 120 kV. XRD measurements were carried out using a PANalytical X’Pert3 MRD diffractometer with monochromatized Cu Kα radiation (λ = 1.54 Å) at 40 kV and 30 mA and were recorded in the range from 10 to 80° (2θ). Raman spectrum was recorded on the XploRA Micro-Raman spectrophotometer (Horiba) with a range between 500 and 4000 cm^−1^. Nitrogen sorption isotherms were measured at −197 °C using a Micromeritics ASAP 2000. Before the experiments, the samples were dried at 120 °C and evacuated for 8 h in flowing argon at the flow rate of 60 standard cubic centimeters per minute at 140 °C. Surface area, pore size, and pore volumes were obtained from isotherms using the conventional Brunauer–Emmet–Teller (BET) and Barrett–Joyner–Halenda (BJH) equations. XPS spectra were achieved using a K-Alpha (Thermo Scientific Seoul korea). CasaXPS software (Pilsworth Road, Heywood) was used for the deconvolution of the high-resolution XPS spectra. All the instrumentation analyses were performed at the core research support center for natural products and medical materials of Yeungnam University.

## 4. Results and Discussion

### 4.1. Analysis of the Structure of the Prepared Materials

Figure 1a shows the Raman spectra of the prepared NRPC materials. The presence of carbon gave two bands, for all the prepared materials, one at 1354 cm^−1^, indicated as ‘D’ band, which is an indication of the degree of disorderness, and the other at 1586 cm^−1^, indicated as ‘G’ band, which arises due to graphitization. The ratio of I_D_ and I_G_ provide the value of ‘R’, indicating the degree of disorderness [22]. As could be observed, NRPC/NiMn gave the highest ‘R’ value showing increased disorderness due to the inclusion of bimetallic compounds into the NRPC framework [23,24]. Moreover, the presence of M-O-M bonds gave their representative peak at 537 cm^−1^ for NRPC/Mn, NRPC/Ni and NRPC/NiMn [25]. Figure 1b presents the XRD patterns of the prepared materials. The two main diffraction planes of carbonaceous materials were observed at 2θ value of 24.2° and 44.6° for all the NRPC materials, corresponding to (002) and (001) planes, respectively. The presence of (002) plane indicated the hexagonal graphitic structure of NRPC carbon materials. Other than the neat NRPC, the remaining NRPC/metallic compounds have an interlayer distance of 7.75 Å indicating d_003_ plane, due to the rhombohedral phase [26,27]. Obviously, all the NRPC materials exhibit high purity, as no other peaks corresponding to crystalline phase could be observed.

The porous nature of the materials and their surface area were investigated using BET analysis. BET analysis showing N_2_ adsorption/desorption isotherms and pore-size distribution (PSD) of the NRPC materials is presented in Figure 1c,d, respectively. The N_2_ adsorption/desorption isotherms display type IV isotherm with H_3_ hysteresis loop in the relative pressure region P/P_0_ > 0.45, for all the NRPC materials [28,29]. This is an indication of presence of mesoporous nature of the materials (Figure 1c). The pore diameter of all the material, as identified by PSD curves, was found to be between 2–50 nm, indicating presence of both micropores and mesopores (Figure 1d), exhibiting a narrower distribution of pore size [30,31,32]. The neat NRPC material exhibits the largest specific surface area of 1192 m^2^/g and pore volume of 0.65 cm^3^/g. Among the prepared materials other than neat NRPC, NRPC/NiMn shows increased specific surface area and pore volume of 365 m^2^/g and 0.42 cm^3^/g, respectively.

XPS analysis was conducted to analyze the chemical state of C, N, O, Mn and Ni elements present in NRPC materials and depicted in Figure 2a–f. The survey spectrum in Figure 2a shows five distinct peaks at 284, 399, 532, 643 and 867 eV, corresponding to the binding energies of C, N, O, Mn and Ni, thereby confirming their presence. We can say that among the five elements, three elements (i.e., C, N and O) originate from polybenzoxazine and the remaining two elements (i.e., Mn and Ni) from bimetallic counterparts. Each peak from the survey spectrum was deconvoluted to obtain a more detailed binding property of the elements. The C1s spectrum in Figure 2b was deconvoluted into five different peaks with binding energies at 286.5, 285.0, 285.6, 287.0 and 287.3 eV, respectively [33], due to C–C, C=C, C–N, C=O and O–C=O groups.

The N1s spectrum in Figure 2c was deconvoluted into three different peaks with binding energies of 399.4, 400.3 and 405.3 eV, corresponding to pyrrolic or pyridinic nitrogen, quaternary nitrogen and oxidized nitrogen, respectively. Figure 2d shows the deconvoluted spectrum of O1s. The spectrum was deconvoluted into two different peaks at 532.4 and 533.2 eV, due to the quinone and phenolic hydroxyl groups, respectively [34,35,36]. The two spin states of Mn and Ni (Figure 2e,f), i.e., Mn 2p_3/2_ and Mn 2p_1/2_ and Ni 2p_3/2_ and Ni 2p_1/2_, were observed at 641.2 and 653.6 eV and 855.7 and 873.7 eV, respectively [37,38,39,40], indicating the presence of Mn^2+^/Mn^3+^ and Ni^2+^/Ni^3+^. All these analyses proves that the synthesized NRPC materials contain nitrogen and oxygen incorporated carbon material along with bimetallic compounds.

The morphology of NRPC, NRPC/Mn, NRPC/Ni and NRPC/NiMn was analyzed using SEM and TEM and is depicted in Figure 3 and Figure 4, respectively. From the SEM images (Figure 3a–c), it is clearly seen that the neat NRPC displays a porous structure as it is composed of only carbon materials, at its maximum. But for NRPC/Mn (Figure 3d–f) and NRPC/Ni (Figure 3g–i), petal-like structure with sharp spikes were found to be deposited upon the carbon surface.

This shows that both Mn and Ni can be inserted into the carbon structure through this process. However, a proper structure was not obtained with single metal inclusion. To our surprise, NRPC/NiMn displays a proper flower-like structure (Figure 3j–l), where carbon forms the base of the flower, upon which bimetallic species form petals that are interconnected with each other to represent a complete three-dimensional hierarchical nanoflower morphology. Moreover, the petals are oriented and united in a uniform manner, and no damaged petals could be found. This type of structure is well suited to act as a proper catalyst and to bring about excellent electrochemical stability. To further study in depth about the morphology of NRPC/NiMn, TEM images were taken and shown in Figure 4a–f. TEM images also display porous framework with spikes being spread throughout the porous framework, indicating that the bimetallic nanoparticles are embedded into the porous carbon structure. The SAED pattern (Figure 4f) confirms the presence of different planes (101/012), (015) and (018) of the bimetallic compounds. This further confirms that both SEM and TEM are in good agreement with each other. EDX mapping (Figure 5a–g) also confirms the presence of C, N, O, Mn and Ni in NRPC/NiMn. This further proves the successful incorporation of N and O in the carbon framework and deposition of bimetallic compounds on the carbon structure. The formed structure is expected to be effective to perform as an electrocatalyst in the water-splitting reaction, which is further evaluated by electrochemical studies [41,42].

### 4.2. Fabrication of Working Electrode

The prepared materials, viz., NRPC, NRPC/Mn NRPC/Ni and NRPC/NiMn were used for the fabrication of the working electrode. The fabrication of working electrode for NRPC is given below.

At first NRPC and PVDF with the wt. % of 95:5 were ground well in an appropriate amount of N-methyl-2-pyrrolidone to obtain a homogeneous paste. The resulting homogeneous paste was coated on the Ni foam with an area of 1 cm^2^ by the drop-casting method and sequentially the electrode was kept at 100 °C in a hot air oven for 48 h to dry the electrodes. In a similar way, the remaining electrodes, i.e., NRPC/Mn, NRPC/Ni and NRPC/NiMn, were also prepared. After successful fabrication, the obtained working electrodes were examined for electrocatalytic activity.

### 4.3. Electrochemical Studies

The electrochemical activity of HC/NiMn towards OER was analyzed in O_2_ saturated 1 M aqueous KOH, employing three-electrode system. Figure 6a displays the polarization curves of the prepared materials towards OER at a scan rate of 5 mV s^−1^. The overpotential value could be obtained from the polarization curves, which is the potential required to reach the current density of 10 mA cm^−2^. The value of overpotential is directly related to 12% efficiency of solar-to-fuel conversion device. Therefore, lower the overpotential value, higher is the efficiency. Until now IrO_2_ and RuO_2_ show excellent performance for OER with an overpotential of 330 and 400 mV, respectively [5,43,44]. The materials, i.e., NRPC, NRPC/Mn, NRPC/Ni and NRPC/NiMn produce their onset potentials at 1 mA cm^−2^ of 370, 245, 240 and 220 mV, and overpotentials at 10 mA cm^−2^ of 670, 510, 400 and 370 mV, respectively. Moreover, Tafel slope obtained from Tafel plots (Figure 6b) was used to study the kinetics of the electrocatalytic reactions and their rate determining step. These Tafel plots could be obtained from the steady-state of the polarization curves. As could be seen from the figure, the Tafel slopes were found to be 134 mV dec^−1^ for NRPC, 103 mV dec^−1^ for NRPC/Mn, 82 mV dec^−1^ for NRPC/Ni and 64 mV dec^−1^ for NRPC/NiMn, displaying good linearity. The value of Tafel slopes were smaller for NRPC/NiMn (64 mV dec^−1^) when compared with IrO_2_ (96 mV dec^−1^) and RuO_2_ (121 mV dec^−1^), demonstrating faster OER reaction kinetics of the prepared catalysts [45,46].

The long-term durability of NRPC/NiMn towards OER performance was evaluated by two different techniques: (i) chronoamperometry at two different current densities (10 and 20 mA cm^−2^) for 20 h (Figure 6c) and (ii) polarization curves for 5000 cycles (Figure 6d). It could be observed from the i-t curves that at a current density of 10 mA cm^−2^, an overpotential of 370 mV, and at a current density of 20 mA cm^−2^, an overpotential of 420 mV were obtained. These values could be maintained for a very long duration of 20 h. Similarly, the polarization curves were also stable even after 5000 cycles, featuring excellent long-term stability [1,4,47].

In a similar way, the electrochemical activity of HC/NiMn towards HER was also analyzed in 1 M aqueous KOH, employing three-electrode system. Figure 7a,b represent the polarization curves towards HER and the Tafel plots obtained from the steady-state polarization curves. The polarization curves display onset potentials of 73, 47, 35 and 30 mV @ 1 mA cm^−2^ and overpotentials of 324, 207, 155 and 136 mV @ 10 mA cm^−2^, respectively, for NRPC, NRPC/Mn, NRPC/Ni and NRPC/NiMn. It could be observed that there is a sharp increase in the cathodic current beyond this value. The obtained overpotential value is smaller than the previously reported values for non-noble metal oxides [NCNT/Ni-NiFe_2_O_4_ (250 mV) and NCNT/Ni-NiMn_2_O_4_ (188 mV)] [2,3]. Tafel slopes obtained from the Tafel plots (Figure 7b) were found to be 119, 98, 84 and 78 mV dec^−1^ for NRPC, NRPC/Mn, NRPC/Ni and NRPC/NiMn, respectively. When compared with the prepared materials, NRPC/NiMn displays smaller Tafel slope of 78 mV dec^−1^, comparable to the reported non-noble metal oxides [NCNT/Ni-NiFe_2_O_4_ (76 mV dec^−1^) and NCNT/Ni-NiMn_2_O_4_ (85 mV dec^−1^)]. It is observed that the catalytic property of NRPC/NiMn is efficient towards OER and HER due to the following reasons: the hierarchical three-dimensional flower-like morphology of NRPC/NiMn can speed up the transport of electrons and ions through increased surface area; the anchoring of these metal oxides on carbon spheres avoids agglomeration of these particles, providing excellent structural stability of NRPC/NiMn; and the inclusion of hetero-atom-doped carbon (NRPC) and bimetallic compounds results in providing more active sites at the interface of the electrode and electrolyte due to variable oxidation and reduction states of the metals and synergistically improves the OER process [48,49,50,51,52,53,54,55].

The electrochemical stability of NRPC/NiMn towards HER performance was measured using chronoamperometry at two different current densities and polarization curves, represented in Figure 7c,d, respectively. A very slight drop in current density was observed at 20 h, for both 10 and 20 mA cm^−2^. Moreover, a stable polarization curve was observed even after 2500 cycles. All these results indicate that the prepared NRPC/NiMn material is suitable for both OER and HER performance, with increased activity and electrochemical stability.

## 5. Conclusions

In this study, we have demonstrated the synthesis of a new electrocatalyst, NRPC/NiMn, that exhibits excellent electrocatalytic performance for hydrogen evolution in water-splitting reactions. The NRPC/NiMn electrocatalyst was prepared using a simple and scalable method, and it is preapred from inexpensive and abundant materials. The nanoflower structure of the catalyst plays an important role in enhancing the electrochemical performance, as it provides more active sites by enlarging the surface area and enables increased contact with the electrolyte. In addition to its excellent performance, the NRPC/NiMn electrocatalyst is also stable and cost-effective. It was able to maintain its activity for over 20 h when used continuously. The morphology of NRPC/NiMn electrocatalyst, as observed by SEM and TEM analyses revealed a uniform distribution of Ni and Mn on the NRPC surface. This uniform distribution is essential for the high performance of the catalyst, as it ensures that there are no areas that are depleted in Ni or Mn. Overall, the NRPC/NiMn electrocatalyst is a promising new candidate for use in water-splitting applications, with an overpotential of 370 mV and a Tafel slope of 64 mV dec^−1^ for OER. Moreover, it also exhibits excellent performance, stability and cost-effectiveness, making it a promising option to be used in energy-related technologies. The NRPC/NiMn electrocatalyst can also be used to develop new types of fuel cells, solar cells and other energy technologies. The results obtained from this study provide new insights in the development of new methods for the production of clean and renewable fuels.

## Data Availability

Not applicable.

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
