# Peer review of "Facile Synthesis of Nitrogen-Rich Porous Carbon/NiMn Hybrids Using Efficient Water-Splitting Reaction"

_polymers, 2023, doi:10.3390/polym15143116_

Round 1

Reviewer 1 Report

In this manuscript, the authors reported the fabrication of nitrogen rich porous carbon (NRPC) along with the inclusion of non-noble metal-based catalyst, adopting a simple and scalable methodology.The research results of this manuscript have certain progress. However, there are some problems in this manuscript, which need to be revised before acceptance. The problems and suggestions for improvement in this manuscript are as follows:

1.       The performance of this catalyst is worse than that of other catalysts cited in the introduction. What are the innovations or advantages of this catalyst? Please describe it in the introduction.

2.       Please carefully check the serial number and title of the picture, especially Figure 1 and Figure 3. Besides, please add a ruler to figure 5a.

3.       Besides the peaks mentioned by the author in Figure 1b, what do some miscellaneous peaks of NRPC/metallic compounds correspond to, especially the peaks around 35°? Why is the (002) peak of NRPC different from the position of the NRPC/metallic compounds peak?

4.       What elements do the unlabeled peaks on the XPS survey spectrum of Figure 2a represent?

5.       What is the diffraction ring corresponding to the SADE pattern in figure 4f? Need further processing by the author.

6.       Why should different scanning rates be used for cyclic voltammetry (CV) measurement?

7.       Figure 6 is exactly the same as Figure 7, which is a serious mistake.

8.       Post-reaction characterization is important for electrocatalysts, such as XPS and SEM, which are necessary to elucidate structural stability.

9.       There are also some articles (e.g., 10.3390/nano12152640; 10.1016/j.jmst.2023.02.050) on the electrochemical behavior of electrocatalytic materials, which can be referred to.

 Moderate editing of English language required

Author Response

Response to reviewers’ comments

In this manuscript, the authors reported the fabrication of nitrogen rich porous carbon (NRPC) along with the inclusion of non-noble metal-based catalyst, adopting a simple and scalable methodology. The research results of this manuscript have certain progress. However, there are some problems in this manuscript, which need to be revised before acceptance. The problems and suggestions for improvement in this manuscript are as follows:

  1. The performance of this catalyst is worse than that of other catalysts cited in the introduction. What are the innovations or advantages of this catalyst? Please describe it in the introduction.

Response: The performance of the prepared catalyst suits better in case of HER when compared with the literature reports. The advantages of this catalysts is that the hetero-atom doping in carbon is made easier with polybenzoxazine origin and the bimetal inclusion through simple hydrothermal method. As per your suggestion, it has been described in the introduction part.

  1. Please carefully check the serial number and title of the picture, especially Figure 1 and Figure 3. Besides, please add a ruler to figure 5a.

Response: Figures 1, 3 and 5a has been modified with proper serial number, title and scale bar.

  1. Besides the peaks mentioned by the author in Figure 1b, what do some miscellaneous peaks of NRPC/metallic compounds correspond to, especially the peaks around 35°? Why is the (002) peak of NRPC different from the position of the NRPC/metallic compounds peak?

Response: The peak at 35° is due to the (311) plane of NiMn. With the inclusion of metals into the carbon structure, the orientation of carbon plane is disturbed and so the (002) plane is shifted slightly for NRPC/metallic compounds.

  1. What elements do the unlabelled peaks on the XPS survey spectrum of Figure 2a represent?

Response: The unlabelled peaks around binding energy of 200 eV in the XPS survey spectrum is due to the C1 2p peak.

  1. What is the diffraction ring corresponding to the SADE pattern in figure 4f? Need further processing by the author.

Response: The diffraction ring corresponding to the SAED pattern in figure 4f is due to (101/012), (015) and (018) planes of bimetallic compounds.

  1. Why should different scanning rates be used for cyclic voltammetry (CV) measurement?

Response: Different scanning rates for CV measurements is used to find the kinetics of the reaction and the stability of the catalysts.

  1. Figure 6 is exactly the same as Figure 7, which is a serious mistake.

Response: We do apologize for this error. Figure 7 has been replaced with the correct figure.

  1. Post-reaction characterization is important for electrocatalysts, such as XPS and SEM, which are necessary to elucidate structural stability.

Response: We performed LSV after measuring the catalyst’s stability for both HER and OER. At this time, we did not characterize XPS or SEM, but in our future work, we assure you to keep up your comment to elucidate the structural stability.

  1. There are also some articles (e.g., 10.3390/nano12152640; 10.1016/j.jmst.2023.02.050) on the electrochemical behavior of electrocatalytic materials, which can be referred to.

Response: The mentioned articles has been included in the references. Kindly refer to references 52 and 55.

We believe that all the comments given by the reviewers’ have been addressed properly.

We hope for the acceptance of the manuscript at its earliest.

Reviewer 2 Report

In this work, the authors reported the fabrication of nitrogen rich porous carbon (NRPC) supporting non-noble metal-based catalyst including Ni and Mn species, and obtained good electrochemical performance towards water splitting. The sample NRPC/NiMn shows an over-potential of 370 mV at a current density of 10 mA cm–2 towards OER. Though the OER performance is good at some degree, while the conclusion is not enough supported by the experimental result. So the manuscript should be carried out a major revision before accepted. The following issues should be considered.

1.        The synergistic effect produced by NRPC and bimetallic compounds increases the active sites at the electrode/electrolyte interface and thus speeds up the OER process. The related discussion should be given in the manuscript.

2.        Fig.7 is same as Fig.6, which is wrong. The legend in Fig.7 show the HER performance, while the Fig. 7 gives the OER performance.

3.        The performance of NRPC/NiMn towards water splitting is suggested to compare with the same type of catalyst reported in documents.

4.        Some related works can be referenced to explain the synergetic catalytic effect (Nanomaterials 2023, 13, 74. Applied Catalysis B: Environmental, 2021, 297, 120474)

Author Response

In this work, the authors reported the fabrication of nitrogen rich porous carbon (NRPC) supporting non-noble metal-based catalyst including Ni and Mn species, and obtained good electrochemical performance towards water splitting. The sample NRPC/NiMn shows an over-potential of 370 mV at a current density of 10 mA cm–2 towards OER. Though the OER performance is good at some degree, while the conclusion is not enough supported by the experimental result. So the manuscript should be carried out a major revision before accepted. The following issues should be considered.

  1. The synergistic effect produced by NRPC and bimetallic compounds increases the active sites at the electrode/electrolyte interface and thus speeds up the OER process. The related discussion should be given in the manuscript.

Response: As mentioned by the reviewer, the sentence has been included in the results part of electrochemical studies.

  1. 7 is same as Fig.6, which is wrong. The legend in Fig.7 show the HER performance, while the Fig. 7 gives the OER performance.

Response: We apologize for inputting wrong figure. Figure 7 has been replaced with the correct figure.

  1. The performance of NRPC/NiMn towards water splitting is suggested to compare with the same type of catalyst reported in documents.

Response: As mentioned by the reviewer, the performance of NRPC/NiMn is compared with similar type of catalyst in the results part of electrochemical studies.

  1. Some related works can be referenced to explain the synergetic catalytic effect (Nanomaterials 2023, 13, 74. Applied Catalysis B: Environmental, 2021, 297, 120474)

Response: As mentioned, the related works are referenced in the manuscript. Kindly refer to references 53 and 54.

We believe that all the comments given by the reviewers’ have been addressed properly.

We hope for the acceptance of the manuscript at its earliest.

Reviewer 3 Report

In this manuscript, Periyasamy et al. reported the fabrication of bimetallic NiMn coupled with nitrogen-rich porous carbon for enhanced electrocatalysis of the water splitting half-reactions, i.e., oxygen evolution (OER) at anode and hydrogen evolution (HER) at cathode. Comparison was made with non-metallic and monometallic counterparts in terms of both the structure formation and electrocatalytic activity. The composite also showed good catalytic stability toward both OER and HER. Overall, the research work has good novelty and the manuscript was well organized. I would in general support the publication at the Polymers journal. However, the current manuscript requires some revision to further improve the quality and clarity. The authors might find the below detailed comments useful.

1. The authors reported catalysis for both OER and HER. But the Abstract only reported performance for the OER.

2. The last paragraph of the Introduction used some abbreviations that were not seen elsewhere in the manuscript, for example, hetero atom doped carbon materials (HC) and bimetallic oxides (BMO). Please revise this part for clarity.

3. In Introduction, the authors also mentioned ORR (Ref. 1) in the penultimate paragraph, which is not relevant to this manuscript.

4. What exactly are the bimetallic species? Metals or metal oxides? There is confusion because the Abstract suggested it could be bimetallic alloys, while the Introduction clearly claimed it was metal oxide. Revision should be made to the text to avoid this confusion.

5. Following above, there is no discussion on this metal or metal oxide phase in the discussion of XRD. The authors claimed that “all the NRPC materials exhibit high purity, as no other peaks corresponding to crystalline phase could be observed.” But as seen from Fig 1b, it is clear that there are additional peaks that the authors did not discuss. For example, what are the 018 and 015 reflections?

6. To appeal to a broader readership, recent works about water electrolysis (e.g., Energy Technology, 2022, 10, 2200573; Small, 2021, 17, 2101573; Small Methods, 2022, 6, 2201099) are recommended to be referenced in the Introduction.

7. Fig 2 e and f, the XPS data fitting needs revision. For Mn 2p data, the Mn species should be clearly identified, and the area ratio should follow 2p3/2:2p1/2= 2:1. For Ni 2p data, there is only one peak for Ni2p1/2 but two for Ni2p3/2.

8. Figure 7 is the same figure as in Figure 6. The authors must have mistakenly placed one more time the Figure 6. Please replace with the right figure.

9. Figure 4, the scale bars are either absent (Fig. 4c) or not clear (Fig. 4a, d-f). Fig. 5a, the scale bar should also be clearly presented.

10 For the section “4.3. Electrochemical studies”, part of the content (the testing methods) should be moved to the “2. Materials and Methods” section.

 Minor editing of English language required (like typos). I would like to leave this to the English editing team.

Author Response

In this manuscript, Periyasamy et al. reported the fabrication of bimetallic NiMn coupled with nitrogen-rich porous carbon for enhanced electrocatalysis of the water splitting half-reactions, i.e., oxygen evolution (OER) at anode and hydrogen evolution (HER) at cathode. Comparison was made with non-metallic and monometallic counterparts in terms of both the structure formation and electrocatalytic activity. The composite also showed good catalytic stability toward both OER and HER. Overall, the research work has good novelty and the manuscript was well organized. I would in general support the publication at the Polymers journal. However, the current manuscript requires some revision to further improve the quality and clarity. The authors might find the below detailed comments useful.

  1. The authors reported catalysis for both OER and HER. But the Abstract only reported performance for the OER.

Response: As suggested by the reviewer, the performance for HER is also included in the abstract part.

  1. The last paragraph of the Introduction used some abbreviations that were not seen elsewhere in the manuscript, for example, hetero atom doped carbon materials (HC) and bimetallic oxides (BMO). Please revise this part for clarity.

Response: The abbreviations corresponding to HC and BMO has been removed and revised accordingly.

  1. In Introduction, the authors also mentioned ORR (Ref. 1) in the penultimate paragraph, which is not relevant to this manuscript.

Response: The performance corresponding to ORR has been removed, as per the reviewer’s suggestion.

  1. What exactly are the bimetallic species? Metals or metal oxides? There is confusion because the Abstract suggested it could be bimetallic alloys, while the Introduction clearly claimed it was metal oxide. Revision should be made to the text to avoid this confusion.

Response: It is bimetal oxides and not bimetal alloys. The words stating bimetallic alloys has been removed in the abstract part.

  1. Following above, there is no discussion on this metal or metal oxide phase in the discussion of XRD. The authors claimed that “all the NRPC materials exhibit high purity, as no other peaks corresponding to crystalline phase could be observed.” But as seen from Fig 1b, it is clear that there are additional peaks that the authors did not discuss. For example, what are the 018 and 015 reflections?

Response: The additional peaks at (015) and (018) are due to the rhombohedral phase of bimetallic compounds.

  1. To appeal to a broader readership, recent works about water electrolysis (e.g., Energy Technology, 2022, 10, 2200573; Small, 2021, 17, 2101573; Small Methods, 2022, 6, 2201099) are recommended to be referenced in the Introduction.

Response: As suggested, the mentioned references has been included in the introduction section. Kindly refer to references 19-21.

  1. Fig 2 e and f, the XPS data fitting needs revision. For Mn 2p data, the Mn species should be clearly identified, and the area ratio should follow 2p3/2:2p1/2= 2:1. For Ni 2p data, there is only one peak for Ni2p1/2 but two for Ni2p3/2.

Response: The deconvoluted XPS peaks are properly fitted for Mn 2p and Ni 2p and the figure 2 e and f has been replaced.

  1. Figure 7 is the same figure as in Figure 6. The authors must have mistakenly placed one more time the Figure 6. Please replace with the right figure.

Response: We apologize for this mistake. Figure 7 has been replaced with correct figure.

  1. Figure 4, the scale bars are either absent (Fig. 4c) or not clear (Fig. 4a, d-f). Fig. 5a, the scale bar should also be clearly presented.

Response: Figures 4a-f and Fig. 5a has been replaced with clear scale bars.

  1. For the section “4.3. Electrochemical studies”, part of the content (the testing methods) should be moved to the “2. Materials and Methods” section.

Response: As suggested, the testing methods in electrochemical studies has been moved to materials and methods section.

We believe that all the comments given by the reviewers’ have been addressed properly.

We hope for the acceptance of the manuscript at its earliest.

Round 2

Reviewer 1 Report

Accept!

Reviewer 2 Report

The authors have answered my questions, and the manuscript is now acceptable.